# Endobronchial Ultrasound/Transbronchial Needle Aspiration-Biopsy for Systematic Mediastinal lymph Node Staging of Non-Small Cell Lung Cancer in Patients Eligible for Surgery: A Prospective Multicenter Study

**DOI:** 10.3390/cancers15164029

**Published:** 2023-08-09

**Authors:** Duilio Divisi, Gabriella Di Leonardo, Massimiliano Venturino, Elisa Scarnecchia, Alessandro Gonfiotti, Domenico Viggiano, Marco Lucchi, Maria Giovanna Mastromarino, Alessandro Bertani, Roberto Crisci

**Affiliations:** 1Department of Life, Health and Environmental Sciences, Thoracic Surgery Unit, University of L’Aquila, 67100 L’Aquila, Italy; 2Department of Thoracic Surgery, Cuneo General Hospital, 12100 Cuneo, Italy; 3Thoracic Surgery Department of Experimental and Clinical Medicine, University of Florence, 50121 Florence, Italy; 4Division of Thoracic Surgery, University Hospital of Pisa, 56124 Pisa, Italy; 5Division of Thoracic Surgery and Lung Transplantation, IRCCS ISMETT-UPMC, 90127 Palermo, Italy

**Keywords:** NSCLC, mediastinal staging, ultrasonography of the mediastinum

## Abstract

**Simple Summary:**

Histological and/or cytological evaluation of the mediastinal lymph nodes is essential for the successful treatment of lung cancer. This study analyzes the role of endobronchial ultrasound (EBUS) in the preoperative staging of non-small cell lung cancer. We carried out a prospective study between December 2019 and December 2022 on 217 lung cancer patients eligible for surgical resection. The lymph nodes biopsied, the number of samples, and the likelihood ratio for positive and for negative outcomes were the variables considered. All patients were discharged from hospital on day one. A downstaging and upstaging were noted in 16 patients (8 and 8, respectively, 7.4%). The sensitivity, specificity, positive and negative predictive value, and diagnostic accuracy were 90%, 90%, 82%, 94%, and 90%, respectively. The likelihood ratio for positive and negative results confirmed cancer when present, excluding it when absent. EBUS is the only minimally invasive and easy procedure for mediastinal staging. The direct visualization of the vessels, especially if posterior to the lymph node, allows for method-checking at every step and makes it safe and effective. Therefore, the endoscopist and the histologist/cytologist must have carried out an adequate learning curve in order not to negatively affect the method.

**Abstract:**

Background: The treatment of lung cancer depends on histological and/or cytological evaluation of the mediastinal lymph nodes. Endobronchial ultrasound/transbronchial needle aspiration-biopsy (EBUS/TBNA-TBNB) is the only minimally invasive technique for a diagnostic exploration of the mediastinum. The aim of this study is to analyze the reliability of EBUS in the preoperative staging of non-small cell lung cancer (NSCLC). Methods: A prospective study was conducted from December 2019 to December 2022 on 217 NSCLC patients, who underwent preoperative mediastinal staging using EBUS/TBNA-TBNB according to the ACCP and ESTS guidelines. The following variables were analyzed in order to define the performance of the endoscopic technique—comparing the final staging of lung cancer after pulmonary resection with the operative histological findings: clinical characteristics, lymph nodes examined, number of samples, and likelihood ratio for positive and negative outcomes. Results: No morbidity or mortality was noted. All patients were discharged from hospital on day one. In 201 patients (92.6%), the preoperative staging using EBUS and the definitive staging deriving from the evaluation of the operative specimen after lung resection were the same; the same number of patients were detected in downstaging and upstaging (8 and 8, 7.4%). The sensitivity, specificity, positive and negative predictive value, and diagnostic accuracy were 90%, 90%, 82%, 94%, and 90%, respectively. The likelihood ratio for positive and negative results was 9 and 0.9, respectively, confirming cancer when present and excluding it when absent. Conclusions: EBUS is the only low-invasive and easy procedure for mediastinal staging. The possibility to check the method in each of its phases—through direct visualization of the vessels regardless of their location in relation to the lymph nodes—makes it safe both for the endoscopist and for the patient. Certainly, the cytologist/histologist and/or operator must have adequate expertise in order not to negatively affect the outcome of the method, although three procedures appear to reduce the impact of the individual professional involved on performance.

## 1. Introduction

Lung cancer treatment is closely related to the stage of the tumor at diagnosis [1,2]. The involvement of the mediastinum radically modifies the therapeutic approach; therefore, it is crucial to carry out a correct preoperative evaluation of the lymph node stations [3]. On the one hand, invasive diagnostic methods of the mediastinum involve high costs and disadvantages, on the other hand, they ensure a wide collection of tissue samples and a histological diagnosis in most cases [4,5,6]; however, today there are alternative methods that have the same diagnostic yield as conventional approaches but with significantly less invasiveness. Endobronchial ultrasound-transbronchial needle aspiration/biopsy (EBUS-TBNA/TBNB) represents a turning point since it allows us to obtain results comparable to more invasive techniques in terms of diagnostic accuracy, sensitivity, and specificity, with minimal discomfort of the patient [7,8,9]. Despite the comforting data from the literature [10,11,12], some authors advocate and re-propose surgical methods as essential in the staging of lung cancer, considering EBUS to be too dependent on the surgeon/endoscopist and on the cytologist and/or pathologist [13,14,15]. The purpose of this study is to highlight the real role of EBUS in the preoperative evaluation of the mediastinum in non-small cell lung cancer (NSCLC) ahead of surgery.

## 2. Method

Two hundred and seventeen non-small cell lung cancer (NSCLC) patients, eligible for surgery, underwent preoperative mediastinal staging using EBUS after total-body computed tomography (CT) scan—with and without contrast enhancement—and/or total body fluorine-18 fluorodeoxyglucose positron emission tomography/computed tomography (18F-FDG-PET/CT). The study was conducted between December 2019 and December 2022, according to the ACCP [16] and ESTS [17] guidelines; the Internal Review Board of the University of L’Aquila approved the prospective research (protocol number: 70302). Furthermore, the lymph nodes examined were chosen on the basis: (1) CT, if the diameter of the lymph nodes was ≥1 cm in the respective minor axis, independently of whether there was uptake on PET/CT; (2) PET/CT, if there was uptake (SUV_max_ > 2.5) in lymph nodes with a diameter < 1 cm in the minor axis on CT. The clinical characteristics of patients are reported in Table 1 and Table 2. The choice between anesthesia (propofol) and moderate sedation, the patient’s decubitus, and the type of needles used depended on each center’s experience and practice. On average, three procedures (range: 2–5) for 50–60 needle passes were performed for each lymph node station sampled.

### 2.1. Primary Endpoints

Analysis of specificity, sensitivity, diagnostic accuracy, positive predictive value, negative predictive value of EBUS.

Evaluation if there is a “Threshold Value” of procedures that positively affects the diagnostic yield of the EBUS.

### 2.2. Secondary Endpoint

Analysis of diagnostic performance of cytology versus histology based on the different types of needles used.

### 2.3. Statistical Analysis

The analysis was performed using SPSS 10.0. Data were entered into a database using SPSS Data Entry II (SPSS, Inc., Chicago, IL, USA). Spearman’s Rank-Order Correlation was used for both dichotomous and continuous variables. The Multiple Regression test allowed us to verify the correlation between early and final staging. All *p* values < 0.05 were considered to indicate significance with a 95% confidence interval.

## 3. Results

No complications during or after the procedure were noted. The mean duration of an EBUS was 27 min (range: 22–32 min) with a length of stay of 1 day. The average number of lymph nodes sampled was 2 hilar and 5 mediastinal nodes. The pathological tumor stage is described in Table 3. The outcomes relating to the primary endpoints are described in Table 4. Upstaging and downstaging were the same (8 vs. 8 patients) and statistically insignificant (*p* = 0.61). The negative predictive value (NPV)—which indicates the probability that a negative lymph node (EBUS test) is really negative at definitive histology—is 94% with a diagnostic accuracy of 90%. The positive likelihood ratio—which indicates the ratio between the probability that a lymph node is positive for both EBUS and definitive histology and the probability that a lymph node negative for EBUS (false negative) is positive for definitive histology—showed a good confidence level (LR+: 9). The inverse ratio is to be considered in the negative likelihood ratio, which also showed a good level of confidence (LR−: 0.9). Three procedures represented the cut-off for improving the diagnostic yield of EBUS. Concerning the secondary endpoint, we did not highlight any statistically significant difference (*p* = 0.78) in the performance of the procedure using the cytology needles or the biopsy needles. Spearman’s Rank-Order Correlation showed a statistically significant correlation between the different paired variables examined (Table 5), validating the analysis performed and the results obtained. Table 6 identifies the regression coefficient between the early stage and final stage (R = 0.877) and displays that 88% of the data did not deviate from the mean (Figure 1; 95% confidence interval; *p* = 0.00001). This indicates that the preoperative evaluation of the mediastinum in EBUS is equivalent to that obtained after surgical resection, testifying to the high degree of reliability of the endoscopic method.

## 4. Discussion

This prospective multicenter study displayed that systematic endosonography in NSCLC patients with a resectable lung tumor is a highly reliable method in the preoperative staging of the mediastinum, being characterized by an overall performance of 90% and a good confidence interval for the likelihood ratio. Our results are in agreement with the experience of Bousema et al. [18], who, after a negative EBUS, carried out surgical resection immediately in 171 patients and, subsequently, a confirmatory mediastinoscopy in 155. Mediastinoscopy determined eight minor (4.6%) and three major (1.7%) complications and an unforeseen reduction in N2 by only 1.03%. The authors concluded that confirmatory mediastinoscopy can be omitted in patients with resectable NSCLC, also reducing delays in treatment. Based on these considerations, the use of video-assisted mediastinoscopic lymphadenectomy (VAMLA) or transcervical extended mediastinal lymphadenectomy (TEMLA) [19,20] does not appear justified due to the continuous search for minimally invasive diagnostic and surgical approaches; hence, the need for a re-evaluation of the indications is imperative in order to tailor the procedures on each patient. This concept is stressed by Mullins et al. [21] regarding early-stage inoperable NSCLC patients, who underwent stereotactic body radiotherapy (SBRT) after CT-guided needle biopsy (Group 1: 79 patients) and navigational bronchoscopy with EBUS for hilar and mediastinal staging (Group 2: 79 patients). The authors, having not found statistically significant differences in the recurrence and survival outcomes between the two groups, concluded that the choice of nodal assessment must be carefully evaluated, especially in patients with borderline clinical conditions. One question seems extremely timely: if the use of EBUS—the only minimally invasive method—must also be weighed on the patient, how can mediastinoscopy be promoted? Rami-Porta et al. [22], in an update on lung cancer staging through a review of the literature, showed a sensitivity for EBUS ranging between 17% and 41% in N0 patients at PET/CT and between 38% and 53% in N1 resectable patients at PET/CT. On the contrary, videomediastinoscopy, presenting a sensitivity between 78% and 97% and a negative predictive value between 83% and 99%, could be considered the preferred method for preoperative mediastinal staging. The authors confirmed this belief in an editorial [23], commenting on the predictive model proposed by Verdial et al. [24] concerning lung cancer staging. This model, through a false-positive rate of 44% and a true-positive rate of 97%, could potentially reduce the use of invasive procedures. In the literature, there is some variability about NPV ranging from 87.7% to 93.4% and sensitivity ranging from 35% to 60% [25,26]; our study revealed a sensitivity of 90% and a negative predictive value of 94%, although the use of cytology needles and biopsy needles did not show any statistically significant improvements in the accuracy of EBUS (*p* = 0.78). This secondary endpoint can be explained, as cytology needles often allow the removal of tissue fragments to be used for histological evaluation. In our experience, three procedures represented the cut-off for improving the diagnostic yield of EBUS, reducing the influence of the operator and the pathologist related to their individual skills, as demonstrated by the linear regression analysis performed on the number of procedures applied for each lymph node station. Our outcomes seem to contradict the findings made by Czarnecka-Kujawa et al. [27] about the difficulty of sampling micrometastases or the inaccessibility of some lymph nodes in the same station, which negatively affects the diagnostic performance of EBUS. In fact, the use of different needles according to the morphological characteristics (i.e., rubbery, necrotic, or hard consistency) of the lymph nodes detected using CT or PET as well as the possibility of visualizing the procedure in each of its phases using ultrasound allows us to adapt the technical option to the anatomy, optimizing the risk/benefit ratio according to the patient. The preference for anesthesia or sedation and the discretion of the type of needle used by each individual center—although this did not affect the homogeneity of the data collected—could represent a bias in this study. Also not to be overlooked is the incidence of SARS-CoV-2 infection, which on the one hand reduced the number of patients who could be enrolled, on the other could have affected mediastinal lymphadenopathy by influencing the diagnostic yield of the EBUS.

## 5. Conclusions

EBUS is the only minimally invasive technique with a high performance index for preoperative staging of lung cancer [28,29,30]. Its greatest advantage is the visualization of the lymph node station in real-time from the front and back, which helps to understand the relationships of the vessels and have a mastery over the depth of the biopsy. Concerning this, mediastinoscopy, guaranteeing only a frontal view, can be burdened by a non-negligible rate of hemorrhage as it fails to evaluate the distance or the infiltration of a vessel posteriorly. Furthermore, in our experience, we never received negative feedback regarding the analysis of biomarkers (EGFR, ALK, ROS, PD-L1, etc.) in biopsies performed through the EBUS; it is obvious that this depends on the quality and suitability of the material sent to the anatomopathologist but it is a problem common to every diagnostic procedure that cannot be exclusively attributed to EBUS.

A prospective international study would be desirable to establish which approach is indicated for the staging of NSCLC patients with the same clinical conditions and comorbidities.

## Figures and Tables

**Figure 1 cancers-15-04029-f001:**
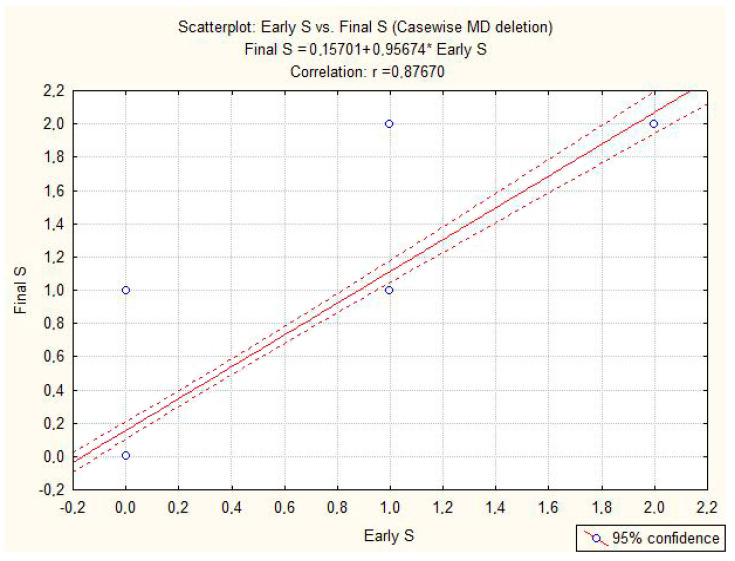
The two dashed curves display the high agreement (R = 0.87670) around the mean.

**Table 1 cancers-15-04029-t001:** Patient demographics and clinical characteristics.

Age				
	Mean ± DS	Min–Max	Median	
	71.7143 ± 8.35	36–87	73	
Sex				
	Male (%)	Female (%)		
	135 (62)	82 (38)		
BMI				
<18.5	31 (14.3)			
18.5–24.9	134 (61.7)			
>25.0	52 (23.9)			
Oncological History				
	YES (%)	NO (%)		
	57 (26)	160 (74)		
Nodal Station Sampling				
		*n* (%)	N1 (%)	N2 (%)
	2L	0 (0)		
	2R	8 (1.5)		
	4L	51 (10)		
	4R	96 (20)		
	5	10 (2)		
	6	5 (1)		
	7	195 (40)		
	10L	30 (6)		
	10R	54 (11)		
	11L	14 (3)		
	11R	28 (5.5)		
	Total number of lymph nodes sampled	491	126 (26)	365 (74)
Primary Tumor Location				
		*n* (%)		*p* value
	Central	45 (21)		
	Periphery	172 (79)		*p* < 0.00001

**Table 2 cancers-15-04029-t002:** Procedures and histological diagnosis and morbidity indices. Br: fiber-optic bronchoscopy; CT-N: CT-guided needle biopsy; EB: EBUS-TBNA-TBNB; Int: intra-operative; Eastern Cooperative Oncology Group (ECOG); Charlson Comorbidity Index (CCI).

Histological Diagnostic Methods					
	*n* (%)				*p* value
Bronchoscopy	31 (14)		Br vs. CT-N		*p* = 0.0001
CT-Needle Biopsy	93 (43)		Br vs. EB		*p* = 0.1613
EBUS	41 (19)		Br vs. Int		*p* = 0.0082
Intra-operative	52 (24)		CT-N vs. EB		*p* = 0.0001
			CT-N vs. Int		*p* = 0.0001
			EB vs. Int		*p* = 0.2056
Tumor Histology					
	*n* (%)				
Squamous	57 (25)				
Adenocarcinoma	143 (66)				
Undiffer. Carc.	3 (2)				
Carcinoid (atypical)	3 (2)				
Other	11 (5)				
	Average Rank	Sum of Ranks	Mean	Std.Dev	*p* value
ECOG 0–5	1.099.078	238.5	0.493088	0.653469	
CHARLSON	1.900.922	412.5	3.470.046	2.325.438	*p* < 0.00001

**Table 3 cancers-15-04029-t003:** Lung cancer stage according to TNM classification, eighth edition.

Pathological Tumor Stage				
		*n* (%)		*p* Value *
		Early Stage	Final Stage	
	pT1a	18 (8)	24 (11)	0.2891
	pT1b	36 (17)	37 (17)	ns
	pT1c	39 (18)	33 (15)	0.2
	pT2a	40 (18)	45 (21)	0.215
	pT2b	27 (12)	21 (10)	0.25
	pT3	38 (18)	41 (19)	0.21
	pT4	19 (9)	16 (6)	0.11
	Total	217	217	
				***p* value ***
**Pathological Nodal Stage**				
	pN0	133 (61)	140 (65)	0.194
	pN1	39 (18)	40 (18)	ns
	pN2	45 (21)	37 (17)	0.144
		217	217	

* *p* values were calculated using Pearson’s Chi-square test.

**Table 4 cancers-15-04029-t004:** Analysis of primary endpoints. The highly positive linear correlation (r = 0.98) indicates that 3 procedures are necessary for the best performance of EBUS. * = Number of Procedure.

Primary Endpoints			
		Early Stage	
		Nodal downstaging	Nodal Upstaging
	N0	7 (3%)	0
	N1	1 (1%)	0
	N2	0	8 (4%)
Accuracy of test			
		Early Stage–Final Stage	
	Sensitivity	90%	
	Specificity	90%	
	PPV	82%	
	NPV	94%	
	Accuracy	90%	
	LR+	9	
	LR−	1	
Number of Procedure *			
	Early Stage	*n* (%)	
2 *	N0	35 (16)	
2 *	N1–N2	15 (7)	
3 *	N0	71 (33)	R = 0.98786
3 *	N1–N2	41 (19)	
4 *	N0	27 (12)	
4 *	N1–N2	28 (13)	
		217	

**Table 5 cancers-15-04029-t005:** Spearman’s correlations show the strength of the relationship between the variables considered, with a highly significant *p*.

Pair of Variables	Valid N	Spearman’s R	T (N-2)	*p*-Level
ECOG:0.5 and CCL	217	0.175368	2.61187	0.009640
ECOG:0.5 and Early S.	217	−0.200989	−3.00846	0.002939
ECOG:0.5 and Final S.	217	−0.215217	−3.23143	0.001425
CCL and Early S.	217	0.316724	4.89615	0.000002
CCL and Final S.	217	0.428564	6.95508	0.000000
Early S. and Final S.	217	0.820610	21.05470	0.000000

Marked correlations are significant at *p* < 0.05.

**Table 6 cancers-15-04029-t006:** Correlation analysis between early stage and final stage. The results are shown graphically in Figure 1.

Dependent Variable	Multiple R	Multiple R^2^	Adjusted R^2^	SS Model	df Model	MS Model	SS Residual	df Residual	MS Residual	F	*p*
Final Stage	0.876699	0.768602	0.767525	87.66309	1	87.66309	26.39221	215	0.122754	714.1335	0.000

## Data Availability

The data can be shared up on request.

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
