# Peer review of "Endobronchial Ultrasound/Transbronchial Needle Aspiration-Biopsy for Systematic Mediastinal lymph Node Staging of Non-Small Cell Lung Cancer in Patients Eligible for Surgery: A Prospective Multicenter Study"

_cancers, 2023, doi:10.3390/cancers15164029_

Round 1

Reviewer 1 Report

There is an ongoing discussion about EBUS vs mediastinoscopy as correct modality for nodal staging and this study is therefore highly relevant.

Abstract: The intro talks about targeted treatment, but study is in preoperative patients where identification of targets for medical treatment is less relevant. Study is actually about nodal staging by EBUS. So suggest to remove wording around targeted treatment

Methods: Largely technical related to EBUS procedure. The study finds good sensitivity and specificity of EBUS staging confirming the high reliability of the procedure.

Discussion flows well and cites relevant literature. There is also a relevant discussion about sampling with cytological vs histological needle biopsies. It is nice to see that newer studies generally shows better performance of EBUS.

The conclusion which is that EBUS is reliable and offer advantages to mediastinoscopy is valid and supported by the study.

The language is generally good but a English language review would improve readability.

My only comment is that the by opening up with a sentence about targeted treatment gives the reader an appetite for something which the study does not deliver namely whether cytology and small bore biopsy from EBUS can be used for reliable biomarker analysis of EGFR, ALK, MET, ROS, PD-L1 etc. I would suggest the authors to look into whether their material can be sued for this analysis because that is considered (at least in the US) a major disadvantage for EBUS for advanced NSCLC.

minor issues to resolve

Author Response

Dear Reviewer,

thank you for your comments.

1) I erased "targeted" in the "Abstract";

2) I added in "Conclusions": 

Furthermore, in our experience we never had negative feedback regarding the analysis of biomarkers (EGFR, ALK, ROS, PD-L1 etc) in biopsies performed through the EBUS; it is obvious that this depends on the quality and suitability of the material sent to the anatomopathologist but it is a problem common to every diagnostic procedure and which cannot be exclusively attributed to EBUS.

Reviewer 2 Report

This a prospective cohort of over 200 NSCLC patients who underwent an EBUS TBNA as part of preopérative staging. 

The sensitivity, specificity, positive and negative predictive value, and diagnostic accuracy were 90%, 90%, 82%, 94% and 90% respectively. 

This is a well-written manuscript reporting a well-designed study. The prospective nature of the study and the clear experience of the investigators are strengths allowing confidence in the results. Reading the manuscript only raised minor queries. 

The nodal status of a subset of patients was down-staged at the time of surgery and others are reported as up-staged. The reviewer supposed that these patients have undergone preoperative chemo (immuno) therapy prior to surgery. Were those patients EBUS evaluated before or after neoadjuvant treatment? Or both?

In the method section it is said that patients underwent  PET - CT as part of the preoperative work-up. It would have been of interest to evaluate the PET performance in terms of sensitivity, specificity, PPV and NPV in comparison with EBUS TBNA. 

The authors should compare the diagnostic performance of EBUS in adenocarcinoma and squamous cell carcinoma apart.

Author Response

Dera Reviewer,

thank you for your considerations.

1) No patients underwent neoadjuvant treatment;

2) You suggest to compare the diagnostic yield of PET versus EBUS and separately evaluating EBUS in adenocarcinoma versus squamous cell carcinoma. This represents the theme of a second article concerning my data, as it is totally beyond the objective of the manuscript. I can totally rewrite the article but I need at least 4 months.